# *Plasmodium falciparum* importation does not sustain malaria transmission in a semi-arid region of Kenya

Christine F. Markwalter[1]*, Diana Menya[2], Amy Wesolowski[3], Daniel Esimit[4], Gilchrist Lokoel[4], Joseph Kipkoech[5], Elizabeth Freedman[6], Kelsey M. Sumner[6,7], Lucy Abel[5], George Ambani[5], Hannah R. Meredith[1], Steve M. Taylor[1,6], Andrew A. Obala[8], Wendy P. O'Meara[1,6]

1 Duke Global Health Institute, Durham, North Carolina, United States of America, 2 School of Public Health, Moi University College of Health Sciences, Eldoret, Kenya, 3 Johns Hopkins Bloomberg School of Public Health, Baltimore, Maryland, United States of America, 4 Department of Health Services and Sanitation, Turkana County, Kenya, 5 Academic Model Providing Access to Healthcare, Eldoret, Kenya, 6 Duke University School of Medicine, Durham, North Carolina, United States of America, 7 University of North Carolina at Chapel Hill, Chapel Hill, North Carolina, United States of America, 8 School of Medicine, Moi University College of Health Sciences, Eldoret, Kenya

* christine.markwalter@duke.edu

**Data Availability Statement:** The sequence data analyzed during the study are available from NCBI (BioProject PRJNA850951). All scripts and

## Abstract

Human movement impacts the spread and transmission of infectious diseases. Recently, a large reservoir of *Plasmodium falciparum* malaria was identified in a semi-arid region of northwestern Kenya historically considered unsuitable for malaria transmission. Understanding the sources and patterns of transmission attributable to human movement would aid in designing and targeting interventions to decrease the unexpectedly high malaria burden in the region. Toward this goal, polymorphic parasite genes (*ama1*, *csp*) in residents and passengers traveling to Central Turkana were genotyped by amplicon deep sequencing. Genotyping and epidemiological data were combined to assess parasite importation. The contribution of travel to malaria transmission was estimated by modelling case reproductive numbers inclusive and exclusive of travelers. *P. falciparum* was detected in 6.7% (127/1891) of inbound passengers, including new haplotypes which were later detected in locally-transmitted infections. Case reproductive numbers approximated 1 and did not change when travelers were removed from transmission networks, suggesting that transmission is not fueled by travel to the region but locally endemic. Thus, malaria is not only prevalent in Central Turkana but also sustained by local transmission. As such, interrupting importation is unlikely to be an effective malaria control strategy on its own, but targeting interventions locally has the potential to drive down transmission.

## Introduction

Increased human mobility influences the spread and transmission of infectious diseases by introducing pathogens into susceptible populations. Human movement has been implicated in outbreaks and changing transmission dynamics for bacterial pathogens such as *Vibrio*

analyses are available on our GitHub (https://github.com/duke-malaria-collaboratory/Turkana_haplotypes).

**Funding:** This work was supported by the National Institute of Allergy and Infectious Diseases (R21AI133013 to WPO, F32AI149950 to CFM). The funders had no role in study design, data collection and analysis, decision to publish, or preparation of the manuscript.

**Competing interests:** The authors have declared that no competing interests exist.

*cholerae* [1, 2], parasites including malaria [3], and viruses such as polio [4], measles [5, 6], dengue [7], Ebola [8], and SARS-CoV-2 [9–11]. In some cases, human movement can increase the pathogen reproductive number, R, and thereby sustain transmission in areas where it would be less than 1 in the absence of importation [12]. Pathogen importation remains a key barrier to achieving elimination, and quantifying its impact on transmission is necessary for designing and implementing successful interventions [13]; however, the impact of pathogen importation is particularly difficult to evaluate in regions with both limited health surveillance capacity and limited contact with remote populations.

In Turkana County, Kenya, human movement and settlement patterns are changing dramatically owing to the influence of oil exploration and resource extraction on economic development [14]. Historically, due to the harsh, semi-arid climate and sparse population of nomadic pastoralists, Turkana County has been considered unsuitable for endemic malaria transmission, resulting in very few malaria control efforts in the region [15]. However, this assumption is challenged by reports of malaria both from routine cases in local health facilities as well as several malaria outbreaks [16, 17]. Consistent with these observations, we recently reported that nearly one-third of the household members of acute malaria patients in Central Turkana were also infected with *Plasmodium falciparum*, the major malaria parasite species in Kenya, suggesting that malaria is endemic in these communities [18]. Given the recent increase in local travel into and out of the region that is concentrated along a limited number of travel routes, it seems plausible that parasite importation, if implicated in sustaining local transmission, could be intervened upon to reduce the burden of malaria. Recently, parasite genotyping has offered valuable insight into parasite population structure on spatial and temporal scales and revealed sources of parasite movement in other settings [19–25]. Therefore, using parasite genotypes to understand the sources and patterns of malaria transmission in Turkana, including those attributable to human movement, could aid in designing and targeting interventions to decrease this unexpectedly high malaria burden in the region.

Here, we investigated the relative contribution of *P. falciparum* parasite importation to local malaria transmission in Central Turkana. To do so, we genotyped *P. falciparum* parasites using amplicon deep sequencing from infected community members and individuals traveling to Central Turkana by bus or plane. We then used these data in concert with self-reported travel histories and malaria transmission modeling to estimate how frequently imported genotypes are incorporated into the local parasite population and understand whether travel fuels malaria transmission. We hypothesized that *P. falciparum* parasites in travelers to the region measurably contribute to and help to sustain local malaria transmission.

## Results

### Study population

From August 2018 to December 2019, we enrolled 1933 people with RDT-positive malaria at six health facilities (index cases), 3353 of their household members, and 1899 inbound passengers to Central Turkana County (S1 Fig). Of these 7185 individuals, we obtained molecular parasite detection results for 7096 (98.8%). Malaria parasites were detected by qPCR in 97.5% (1845/1891) of index patients, 30.7% (1018/3314) of household members, and 6.7% (127/1891) of inbound passengers. Of these 2990 PCR-positive samples, amplification and sequencing were successful for *P. falciparum* circumsporozoite protein (*csp*) in 2521 (84.3%) and for apical membrane antigen 1 (*ama1*) in 2514 (84.1%) samples, across which we identified 72 unique *csp* haplotypes and 88 unique *ama1* haplotypes (Fig 1).

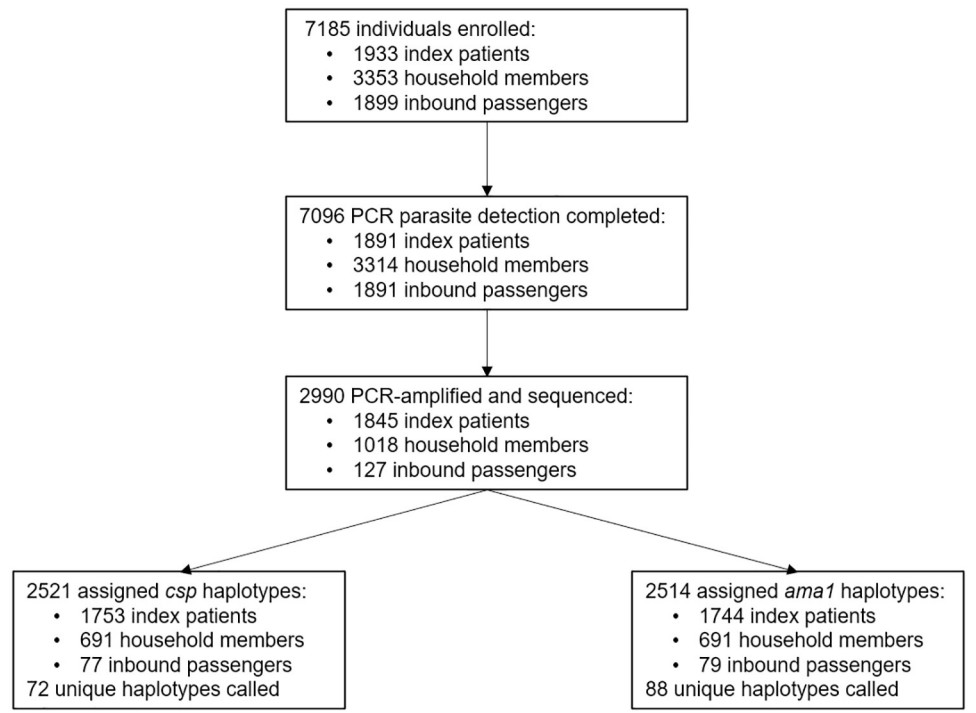

**Fig 1. Study design, enrollment, sequencing, and haplotypes.**

## Recent travel among participants

Among the 7096 participants with parasite detection results, 2048 (28.9%) reported travel in the 2 months prior to enrollment, including all 1891 inbound passengers and 157 community members. Compared to non-travelers, these individuals tended to be 16 years or older (90.1%; 1796/2048) and male (63.5%; 1300/2048) (both p < 0.001). Among community participants, travel in the 2 months prior to enrollment was reported by 95/1891 (5.0%) index patients and 62/3314 (1.9%) household members. Of the community participants reporting travel, 80% (126/157) were enrolled at urban health facilities (67 index cases, 59 household members) within Lodwar town (Table 1). The 157 community-enrolled travelers reported 167 trips, 106 (67.5%) of which were within Turkana County. The majority of trips taken by community members outside of Turkana County were to counties with higher parasite prevalence based on Malaria Atlas Project (MAP) estimates (*Pf* PR$_{2-10}$ 2018–2019 [26]) (26/38 (68%) index case trips, 16/23 (70%) household member trips) (S2 Fig). Inbound passengers were nearly evenly split between residents returning to Central Turkana (n = 948) and visitors (n = 941). Most inbound visitors resided in counties with higher *Pf* PR$_{2-10}$ than Turkana County (70%, 644/926). Inbound passengers reported more trips (median 2, IQR 1–2) than community participants who reported travel (median 1, IQR 1–1) (p < 0.001). Of the places visited by inbound passengers, 47% (828/1768) were to counties with higher *Pf* PR$_{2-10}$ than Turkana County.

## Parasite prevalence in travelers

By RDT results, *P. falciparum* prevalence was lower in inbound passengers (2.3% [44/1891]) than in household members (11.9% [394/3314]) (p < 0.001). A similar difference was observed based on PCR results (S1 Table). Among household members, there was no difference between individuals who did and did not report travel in parasite prevalence by RDT (Fisher Exact

**Table 1. Baseline characteristics based on travel reported.**

| | Overall | Inbound passengers | Community Participants | | |
| --- | --- | --- | --- | --- | --- |
| | | | No Travel | Travel | p-value[2] |
| | n = 7,096 | n = 1,891 | n = 5,048[1] | n = 157[1] | |
| Case type | | | | | <0.001 |
| Index cases | 1,891 | | 1,796 (95%) | 95 (5.0%) | |
| Household members | 3,314 | | 3,252 (98%) | 62 (1.9%) | |
| Inbound passengers | 1,891 | 1,891 (100%) | | | |
| Returning | 948 | 948 (100%) | | | |
| Visitor | 941 | 941 (100%) | | | |
| Age | | | | | <0.001 |
| 15 or younger | 2,864 | 143 (5.0%) | 2,675 (93%) | 46 (1.6%) | |
| 16–40 | 3,293 | 1,295 (39%) | 1,905 (58%) | 93 (2.8%) | |
| Older than 40 years | 891 | 441 (49%) | 433 (49%) | 17 (2.0%) | |
| Gender | | | | | 0.7 |
| Male | 3,510 | 1,228 (35%) | 2,210 (63%) | 72 (2.0%) | |
| Associated Health Facility | | | | | <0.001 |
| Rural | 3,217 | | 3,186 (99%) | 31 (1.0%) | |
| Kerio | 1,023 | | 1,017 (99%) | 6 (0.6%) | |
| Nadoto | 1,490 | | 1,470 (99%) | 20 (1.3%) | |
| Nakechichok | 704 | | 699 (99%) | 5 (0.7%) | |
| Urban | 1,988 | | 1,862 (94%) | 126 (6.3%) | |
| Ngiitakito | 737 | | 723 (98%) | 14 (1.9%) | |
| St. Monica | 732 | | 702 (96%) | 30 (4.1%) | |
| St. Patrick | 519 | | 437 (84%) | 82 (16%) | |
| *P. falciparum* RDT positive | 2,329 | 44 (1.9%) | 2,183 (94%) | 102 (4.4%) | <0.001 |
| *P. falciparum* PCR positive | 2,990 | 127 (4.2%) | 2,753 (92%) | 110 (3.7%) | <0.001 |
| Reporting any symptoms | 2817 | 532 (19%) | 2,165 (77%) | 120 (4.3%) | <0.001 |

[1]n (%).

[2]Pearson's Chi-squared test comparing community participants who did and did not travel.

Test, p = 1) or PCR (p = 0.3304) (Fig 2a). Parasite densities for participants who did and did not report recent travel are shown in Fig 2b. Among actively-detected cases (inbound passengers and household members), those reporting travel harbored slightly lower parasite densities (median 0.27 p/μl, IQR: 0.05–1.55 p/μl) than those who did not report travel (median 1.08 p/μl, IQR: 0.02–13.78 p/μl) (Wilcox test, p < 0.001). The median multiplicity of infection (MOI) as measured by *csp* or *ama1* was 1 for both travelers and non-travelers (Fig 2c). *P. falciparum* prevalence, parasite densities, and MOIs stratified by case type (index cases, household members, and inbound passengers) are shown in S3 Fig.

## Haplotype distribution in Central Turkana

Community participants presenting with malaria were enrolled at three urban health facilities (Ngiitakito, St. Monica, and St. Patrick) located within the town of Lodwar and three rural health facilities (Kerio, Nadoto, Nakechichok) located along the Turkwel river that connects Lodwar to Lake Turkana over approximately 60 km [18].

We first compared haplotype distributions by travel reported (and whether a traveler was visiting Turkana or returning), case type, and urbanicity for individuals residing in the study

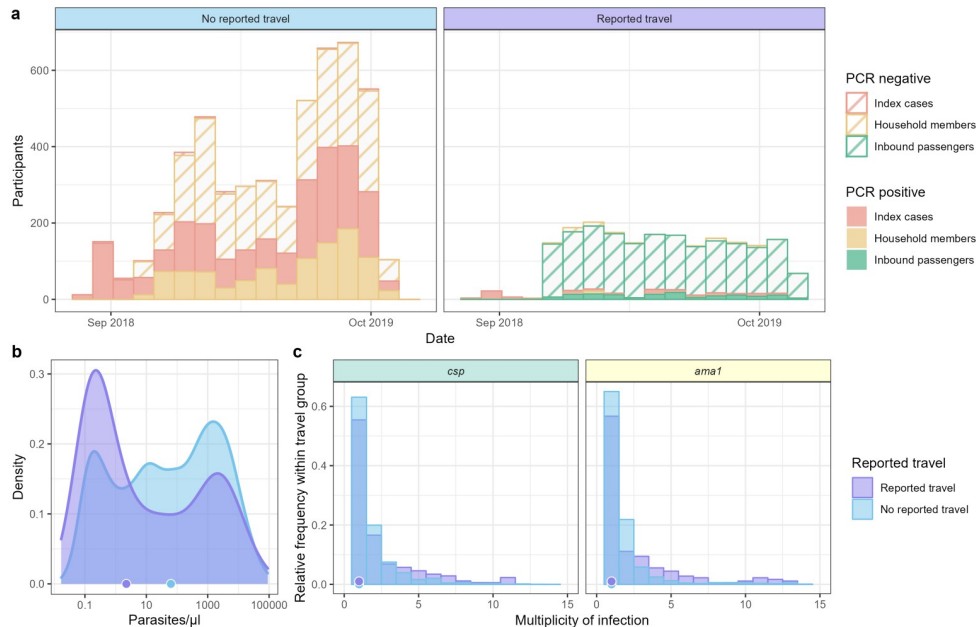

**Fig 2. *Plasmodium falciparum* cases, parasite density, and multiplicity of infection in index cases, household members, and inbound passengers, stratified by reported travel.** (a) *P. falciparum* PCR-positive (solid) and negative (striped) participants by reported travel and case type throughout the study. (b) Parasite density distribution by reported travel. Points on x-axis represent median parasitemias for each group. (c) Multiplicity of infection determined by *csp* (left) and *ama1* (right) haplotype counts in each individual. Points on x-axis represent median MOIs for each case type.

area (S4 Fig). For *csp*, we found a majority (43/72; 60%) of haplotypes were observed in both people who did and who did not report travel; only 4/72 (5.6%) *csp* haplotypes were unique to participants reporting travel, 3 of which were unique to visitors. Similarly, for *ama1*, 45/88 (51%) haplotypes were observed in both individuals who traveled and those who did not, and 7/88 (8%) appeared only in individuals reporting travel, 4 of which were unique to visitors.

Fig 3a and S5–S7 Figs show the proportion of each haplotype observed at each health facility among community participants. For *csp*, a majority of haplotypes detected in community members (51/69; 74%) were detected in samples from multiple health facilities. Similarly, for *ama1*, 59/83 (71%) haplotypes were detected in samples from more than one health facility. Collectively, these observations suggest that *P. falciparum* parasites in Central Turkana circulate across a large area without apparent geographic structuring despite the arid climate and low human population.

## Local pairwise haplotype sharing

We next sought to further understand local malaria transmission by determining whether haplotypes clustered in time or space based on binary pairwise haplotype sharing (i.e. whether a pair of infections share one or more haplotypes). The odds of individuals sharing at least one haplotype decreased by 2.5% with each 2-week interval increase in time between infections for *csp* (Fig 3b, OR 0.9753 95% CI 0.9749–0.9756) and by 1.5% for *ama1* (S5b Fig OR 0.9845 95% CI 0.9842–0.9849). We also observed very small, but statistically significant declines in pairwise haplotype sharing with increasing distance (km) between infections defined at the facility level for *csp* (Fig 3c; OR 0.9986 95% CI 0.9985–0.9987) and *ama1* (S5c Fig; OR 0.9964 95% CI 0.9963–0.9965). Amongst participants with household members who were also infected,

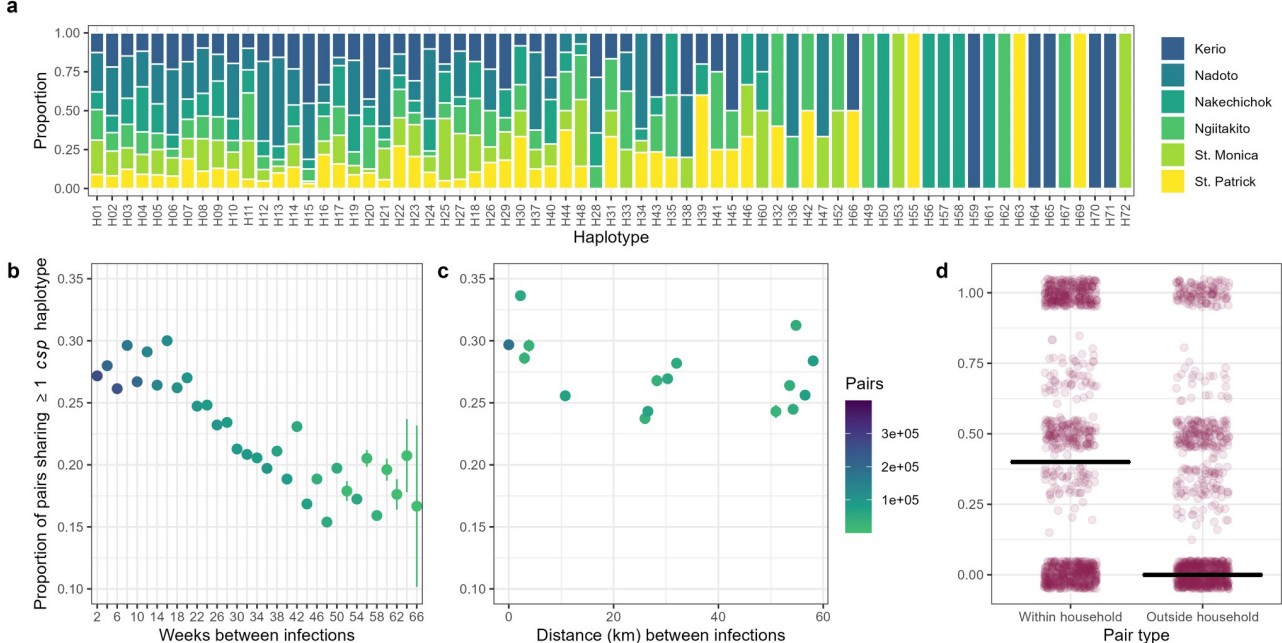

**Fig 3. *csp* haplotype distribution and sharing in Central Turkana.** (a) A majority of *csp* haplotypes were detected at multiple sites across Central Turkana. Columns are individual *csp* haplotypes, and colors indicate the proportion of samples harboring the haplotypes detected at each of 6 enrollment sites. (b) The proportion of infection pairs sharing at least one *csp* haplotype decreases with increasing time between infections. Each dot represents the proportion of all pairs separated by the indicated interval (with n = the color) that shared at least 1 *csp* haplotype. (c) A marginal decrease in the proportion of individuals sharing at least one *csp* haplotype with increasing distance (facility level) between infections was observed. Each dot represents the proportion of all pairs separated by the indicated geographic distance (n = the color) that shared at least 1 *csp* haplotype. (d) The median proportion of individuals sharing at least one haplotype was greater within the household than outside the household. Within household, each dot represents the proportion of household members with whom an individual shared at least one haplotype. Outside household, each dot represents the median proportion of subsampled (n = number of household members, reps = 1000) infection pairs outside the household within ±60 days that share at least one haplotype.

people were much more likely to share a *csp* or an *ama1* haplotype with an infected household member than with an infected person in another household (p < 0.001 for each) (Fig 3d and S5d Fig). These observations suggest that parasite genotypes cluster by household but are generally otherwise more strongly structured temporally rather than geographically.

## Haplotype importation into Central Turkana

To estimate parasite importation into Central Turkana, we first defined a candidate imported haplotype as one that was observed: (1) first in an individual reporting travel and (2) subsequently in individuals who did not report travel outside Central Turkana. This definition identified 7/71 (9.86%) *csp* haplotypes and 11/85 (12.9%) *ama1* haplotypes as candidates for importation. These 18 candidate imported haplotypes were observed in 10 people reporting recent travel, with 1 individual harboring 5 haplotypes new to the study area (Fig 4a), 1 individual harboring 3, and 2 individuals each harboring 2 (S8 Fig, S2 Table). In order to assess whether these candidate imported haplotypes were artifacts of the study time period, we compared the proportions of haplotypes meeting the candidate definition to a null distribution generated by permuting individuals' travel histories (binary exposure). In this analysis, the proportions of *csp* and *ama1* haplotypes meeting importation criteria were compared to permuted null distributions (p = 0.002 for *csp*, p < 0.001 for *ama1*), (Fig 4b and 4c), indicating

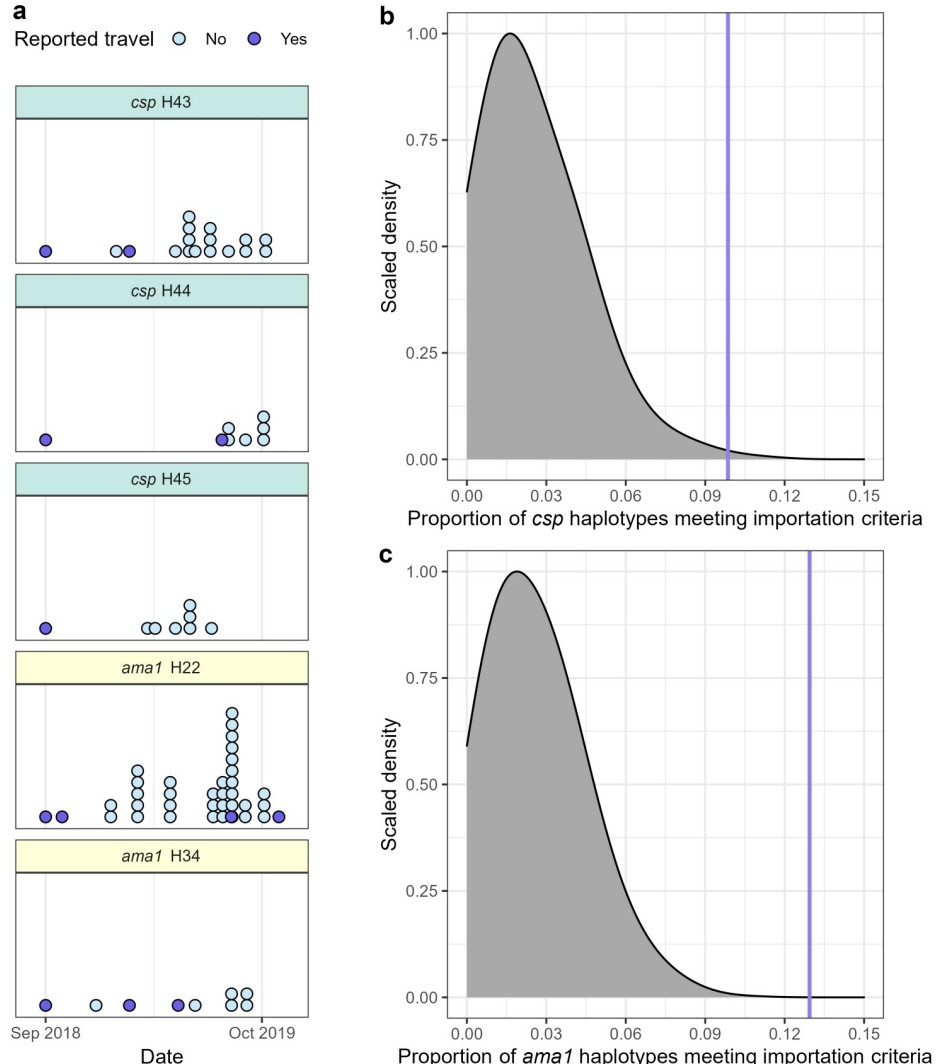

**Fig 4. Evidence for haplotype importation into Central Turkana by individuals reporting travel.** (a) Gray distribution indicates the expected null distribution of *csp* haplotypes meeting the candidate importation criteria, and vertical purple line the observed proportion. The proportion of *csp* haplotypes potentially imported by individuals reporting travel (9.86%) was enriched relative to the permuted null distribution (p = 0.002). (b) Gray distribution indicates the expected null distribution of *ama1* haplotypes meeting the candidate importation criteria, and vertical purple line the observed proportion. The proportion of *ama1* haplotypes potentially imported by individuals reporting travel (12.9%) was also enriched relative to the permuted null distribution (p < 0.001). (c) 5 representative candidate imported haplotypes, all of which were potentially imported by one index case at the St. Patrick health facility who reported travel to western Turkana. Each dot represents one infection with the specified haplotype.

that these candidate imported haplotypes were more common than would be expected under the null.

## Contribution of travelers to *P. falciparum* transmission in Central Turkana

Given the enrichment of putatively imported haplotypes among people reporting recent travel, we rendered transmission networks of all infected individuals sharing at least one *csp* or *ama1* haplotype to visualize the extent of connectivity between people harboring the candidate

imported haplotypes (Fig 5a). In these networks, we observed that these haplotypes comprised a small portion of the total haplotype-based network, suggesting limited contribution to overall transmission. To quantify this, we then compared estimates of the reproductive number (R) in networks with and without people reporting travel in order to estimate the potential contribution of imported parasites to local transmission networks. We used a modified Wallinga Teunis model that incorporated time and distance into transmission weights for all possible infection pairs (S9 Fig) and incorporated genotype information with increasing stringency under the assumption that infections which include the same haplotypes were more likely to be part of the same transmission network. We included in this analysis only PCR-positive community participants for whom we had PCR data for all members of the household (1014 index cases, 978 household members) as well as inbound passengers who were RDT-negative (not treated) and PCR-positive (n = 98) (S10 Fig). In networks of PCR-positive infections, the estimated R was similar when computed with (R = 1.028; 95% CI: 0.979–1.079) or without (R = 1.019; 95% CI: 0.975–1.068) individuals who reported travel in the two months prior to enrollment (Fig 5b). In transmission networks of people with shared parasite genotypes, R based on sharing at least one *csp* haplotype was similar between networks with (R = 1.057; 95% CI: 0.979–1.137) and without (R = 1.049; 95% CI: 0.969–1.138) travelers, with similar results for *ama1* (R = 1.053, 95% CI: 0.969–1.144 among all individuals; R = 1.046, 95% CI: 0.953–1.134 among only non-travelers) (Fig 5b). Concordance between *csp* and *ama1* for pairwise haplotype sharing was moderate (Cohen's Kappa 0.257). Finally, we used this same approach to estimate potential travel effects on individual haplotypes by estimating R for each haplotype among only individuals who shared a specific *csp* or *ama1* haplotype. As shown in Fig 5c, across *csp* or *ama1* haplotypes, there were no differences in R estimates inclusive and exclusive of individuals reporting travel (All p = 1). Collectively, the absence of discernable impact of the exclusion of potential importation to the estimates based on all PCR-positive people, those sharing haplotypes, or on specific haplotypes, suggest that *P. falciparum* parasites detected in recent travelers do not contribute substantially to the sustenance of malaria transmission in the study area.

## Discussion

We collected *P. falciparum* parasites and travel histories in a community-based study in the central region of Turkana County, Kenya in order to investigate the population structure of local parasites and the contribution of parasite importation to local transmission. Using high-resolution parasite genotyping and transmission modelling, we leveraged haplotype diversity to detect signatures of parasite importation into Central Turkana by travelers. Using combined genetic and epidemiological data to model case reproductive numbers, we estimated that parasites carried by travelers contribute to local transmission networks. However, these potential parasite importations are not critical to sustaining malaria transmission in Turkana, which appears rather to suffer from established transmission. Taken together, these results suggest that interrupting malaria importation into the region is unlikely to be an effective malaria control strategy on its own, highlighting the importance for communicable disease control strategies to account for how human movement impacts disease transmission.

*P. falciparum* parasites were harbored by travelers to Central Turkana, including inbound passengers and locally-enrolled participants reporting travel, often from locations with higher *P. falciparum* rates than Turkana County. Importation of pathogens can enhance local transmission generally by either serving as origins of new chains of transmission or more specifically by introducing new genetic variants to which local hosts have not developed immunity [27]. Parasite genotyping revealed that some of these imported infections included haplotypes

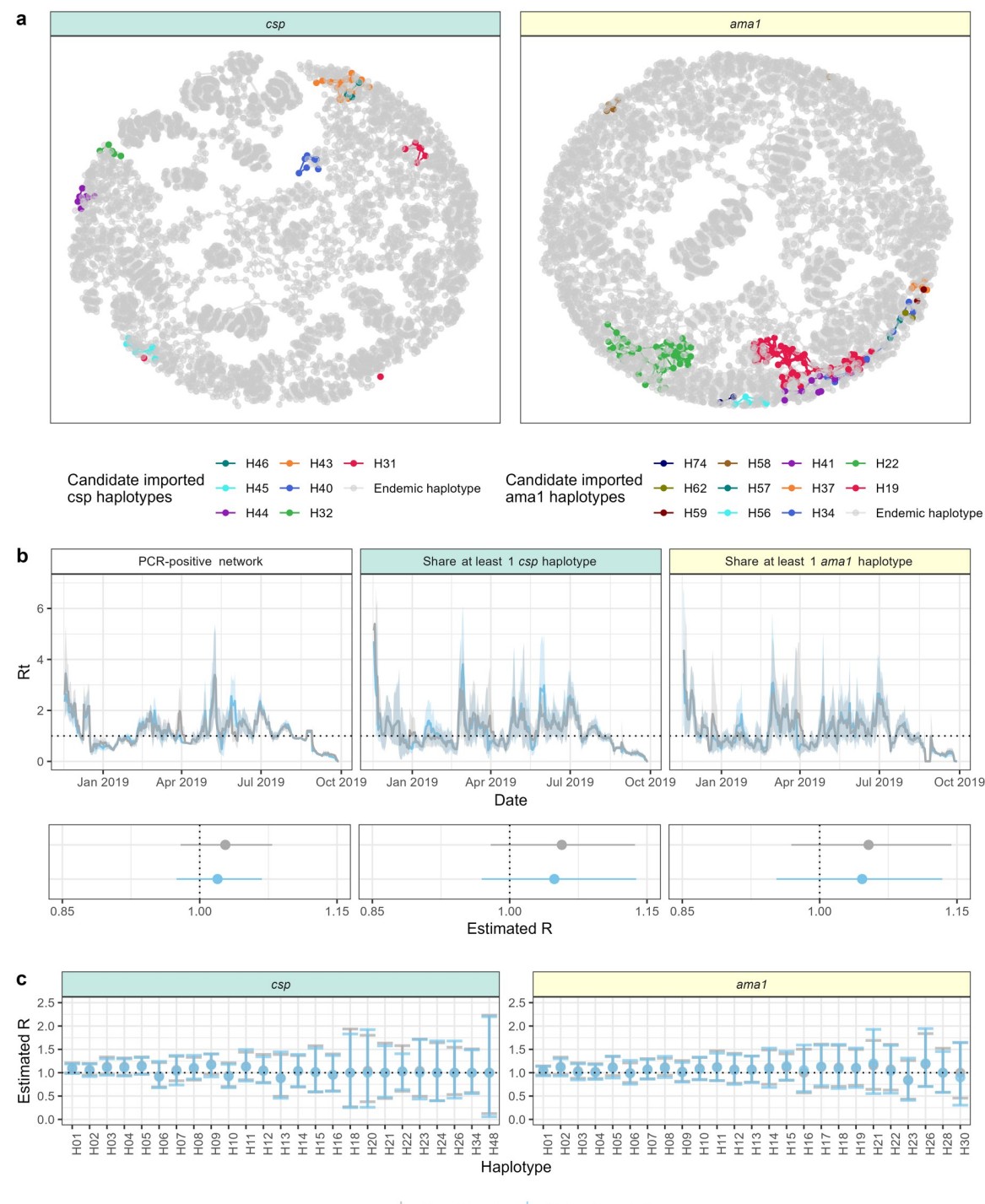

**Fig 5. Contribution of travel to local transmission networks.** (a) Transmission nodes (points) and edges (lines) for individual *csp* and *ama1* haplotypes. Colors represent transmission attributable to candidate imported haplotypes. Edges drawn only if the relative transmission likelihood (Eq 4 in methods) was > 0.02. (b) Top: Weekly estimates of the *P. falciparum* reproductive number (Rt) for transmission networks composed of (left) PCR-positive individuals, (middle) participants sharing ≥ 1 *csp* haplotype, and (right) participants sharing ≥ 1 *ama1* haplotype. Lines indicate population with (gray) or without (blue) participants reporting travel. Bottom: Overall estimates of *P. falciparum* reproductive number in each population both with (gray) and without (blue) the participants reporting travel. Dots indicate point estimate and bars the 95% confidence interval. (c) Haplotype-specific estimates of R for *P. falciparum* transmission networks with (gray) and without (blue) individuals reporting travel for (left) *csp* and (right) *ama1*. Dots indicate point estimate and bars the 95% confidence interval.

first identified in individuals reporting travel and later identified in locally-enrolled individuals who had not traveled outside the study area. Furthermore, several of these *csp* and *ama1* haplotypes were carried into Central Turkana by the same individuals, which supports the idea that these individuals may have acquired their infections in settings with parasite populations distinct from that in Central Turkana. Taken together, these observations suggest that travelers brought new parasites into the study area that spread throughout the local population by establishing local transmission chains.

Importantly, however, infections with these candidate imported haplotypes represented a small proportion of overall transmission in the region (Fig 5a). Indeed, transmission models revealed that effective case reproductive numbers (R and Rt) did not change when travelers were removed from transmission networks. This limited contribution to local transmission could be a consequence of both the relatively low frequency of travel among community-enrolled participants as well as the low prevalence of infection among travelers, both of which reduce the likelihood of establishing new transmission chains. An additional contributing factor to the lack of contribution by imported parasites to local transmission is the unexpectedly high prevalence of infections among the local population independent of travel, which for parasitologic, immunologic, and clinical reasons could attenuate propagation of imported parasites. Collectively, our analyses suggest that transmission is not fueled by travel to the region, but rather occurring locally within Central Turkana.

Several observations support the notion that *P. falciparum* is endemic in Central Turkana despite the low population density and harsh climate that should render it relatively unsuitable. First, across transmission networks constructed with varying genetic relatedness between infections, R estimates approximated 1 over an 11-month period, suggesting sustained but endemic transmission. Secondly, we observed a high parasite prevalence among household members who did not report recent travel (30.8%, 1003/3252), indicating a large reservoir of transmitted and transmissible parasites and apparently well-mixed parasite population between health facilities across the study area. Because these cases were identified using a reactive study design, malaria prevalence in the general community may be lower. To better understand the local *P. falciparum* epidemiology and transmission patterns, we evaluated spatio-temporal haplotype structure in the study area as well as haplotype sharing within households. We observed significant haplotype sharing within households, suggesting that household members are likely participating in the same local transmission network. Finally, we observed patterns of haplotype population structure consistent with a high transmission setting in Western Kenya [28], including haplotype sharing that is highly structured temporally but not geographically beyond the household level. The establishment of endemic malaria in the atypical setting of Turkana County highlights the need for innovative measures to control malaria that are tailored to local conditions.

Our study has several limitations. First, it is possible that some of the identified putatively imported haplotypes may have occurred within the population prior to when we observed them during the study time period. However, the proportion of candidate imported haplotypes was enriched compared to random chance, indicating that the putative haplotype importation was not purely an artifact of the study period. Second, transmission modeling can be sensitive to incomplete observations [29], and though this is mitigated if the fraction of infections observed is constant over time [29, 30], it was unclear how a reactive sampling format would potentially impact our estimated Rt. However, we simulated reactive case detection (RCD) in a previously published longitudinal cohort [31] and found no systematic trends when monthly RCD were restricted to a similar monthly number as our study in Central Turkana (S11 Fig), suggesting that our estimates of R and Rt in Turkana are unlikely to be biased downward by the RCD study format. Third, owing to logistical constraints, inbound passengers from only

one of two air departure points to Lodwar Town were screened for *P. falciparum* malaria. Finally, we may have incompletely cataloged parasite haplotypes owing to the preponderance of low-density infections, which could render our approach insensitive to importations. However, our genotyping method is sensitive to low-density variants; we implemented additional steps to enhance yield in low-density infections, and ultimately obtained *csp* and/or *ama1* genotypes from > 84% of detected infections.

Like other non-traditional settings adjacent to malaria hyperendemic regions, Turkana County has historically been outside the malaria risk map in Kenya and therefore overlooked for country-wide interventions and control measures [15]. Such preventive measures typically include distributions of insecticide-treated bed nets and vector control with indoor residual spraying or other approaches. These are rare in Turkana County [18], though their effectiveness may be attenuated by the unique ecological and epidemiologic features of the region, which include little surface water with only evanescent vector breeding sites, a (semi-)nomadic population, and limited opportunities for bed net use. Among the potential measures for transmission reduction would be prevention of importation, but our finding that malaria is not only prevalent but also sustained by local transmission suggests that interrupting importation is not likely to be an effective malaria control strategy on its own. Spatio-temporal and household-based genetic relatedness of infections, combined with understanding of the unique ecology of the region, can be used to test which interventions may be most effective at reducing transmission in Turkana County.

The deconvolution of imported from locally-transmitted infections is critical in areas receptive to new pathogens as well as in settings that are nearing disease elimination. The methods used in this study to synthesize pathogen genetic data, epidemiological data, and transmission modeling could be generalized to account for human population dynamics in the spread of a broad range of infectious diseases.

## Materials and methods

### Ethics

Written informed consent was provided by all adults and by parents or guardians for individuals under 18 years old. Additional verbal assent was obtained from individuals between 8 and 18 years old. This study was approved by the ethical review boards of Moi University (IREC/2018/74, IREC/2018/191) and Duke University (Pro00100003). Additional information regarding the ethical, cultural, and scientific considerations specific to inclusivity in global research is included in the Supporting Information (S1 Checklist).

### Study area and design

Turkana is a large (68,233 km$^2$), semi-arid county in northwestern Kenya sparsely populated with predominantly semi-nomadic and nomadic people (2019 Turkana County population 926,976) [32]. Lodwar, located in Central Turkana, is the largest town (2019 population 16,931) in the county with a more urbanized and settled population [32]. The main routes into this remote county go through Lodwar by road and air. As previously reported [18], participants presenting with malaria at three urban (Ngiitakito, St. Monica, and St. Patrick) and three rural (Kerio, Nadoto, Nakechichok) health facilities in central Turkana from August 2018 to October 2019 were enrolled along with their household members. The enrollment clinics were selected to represent a broad range of transmission settings around Lodwar.

The Kitale bus terminal serves as the departure point for all road travel to Lodwar, with an estimated volume of 500–1000 passengers to Lodwar per month. Passengers traveling to Lodwar by bus from Kitale (Trans Nzoia County) were enrolled and screened for *P. falciparum*

malaria by RDT at the point of departure. Enrollment at the Kitale bus terminal occurred on 6 days each month from December 2018 to December 2019, and all travelers to Lodwar on the bus or shuttle lines servicing the route on those days were enrolled. Our sampling strategy was highly representative of road travelers; only individuals traveling by private vehicle or lorries, which are relatively rare, were missed. Air passengers departing Eldoret for Turkana were enrolled every Friday at the Eldoret airport from July 2019 to December 2019, which is one of only 2 departure points to Lodwar for air travelers. Enrolled passengers were screened for *P. falciparum* malaria by RDT and provided dried blood spots (DBS) as well as travel and health histories. All RDT-positive participants were provided with Artemether-Lumefantrine treatment (unsupervised), consistent with local guidelines for treatment.

## Parasite detection and genotyping

Molecular detection of *P. falciparum* on DBS from inbound passengers was performed as previously described for community samples [18] using a duplex qPCR assay targeting *P. falciparum pfr364* and human beta-tubulin. Parasite densities were determined based on a standard curve of mock DBS constructed from dilutions of *P. falciparum* 3D7 in whole blood ranging from 0.1 parasites/µl to 2000 parasites/µl. *P. falciparum*-positive samples from both inbound passengers and community members were genotyped across variable segments of the *csp* and *ama1* genes as previously described [31] with the following additions to enhance yield from low-density samples: genomic DNA (gDNA) was extracted from DBS samples using a Chelex-100 protocol that included an overnight incubation with 0.2mg/ml Proteinase K at 56˚C prior to boiling with Chelex. *P. falciparum*-positive gDNA extracts with Ct ≥ 34 were concentrated with an RNA Clean & Concentrator-5 kit (Zymo Research) prior to genotyping. For library preparation, PCR1 reactions included 7 µl of template gDNA when extract Ct was < 28, 18 µl when 28 ≤ Ct < 34, and 15 µl concentrated extract when Ct ≥ 34. PCR2 reactions contained 1.5 µl template when extract Ct was < 28 and 3 µl when extract Ct ≥ 28. Pooled, dual-indexed libraries for *ama1* and *csp* were sequenced on an Illumina MiSeq platform. Haplotypes were inferred as previously described [31]. Briefly, reads were quality-filtered based on length and Phred Quality Score (<15) and mapped to 3D7 reference sequences [33–37]. DADA2 was used for haplotype inference and further read quality filtering, and false discovery was limited by filtering haplotypes as previously described [31, 38]. The final output for downstream analysis was a catalog of unique *ama1* and *csp* haplotypes detected in each individual.

## Local haplotype sharing metrics

We measured haplotype clustering in time, space, and within households using a binary pairwise haplotype sharing metric [28]. Only infection pairs of community members were considered. Analyses were performed separately for *csp* and *ama1*. The effect of the time interval between infections on the odds of sharing a haplotype was measured by logistic regression. Distance between infection pairs was determined as the Haversine distance in km between health facilities to which index cases reported (for index and household members) or inbound passenger destinations. Distance-based haplotype sharing analyses were limited to infection pairs occurring within 60 days of each other. The effect of distance between infections on the odds of sharing a haplotype was measured by logistic regression. For household clustering, the proportion of household members with whom an individual shared at least one haplotype was determined for all participants with at least 2 infected individuals in a household with inferred haplotypes. An analogous proportion for pairs outside the household was determined as the median proportion of subsampled (n = number of household members, reps = 1000) infection pairs outside the household within ±60 days sharing at least one haplotype. A Wilcoxon

Signed-Rank test was used to determine whether there was a difference in haplotype sharing within and outside the household.

### Haplotype importation analysis

Among parasites detected in community members, the definition of candidate imported parasite haplotypes was any *ama1* or *csp* haplotype that was (1) first detected in individuals reporting travel and (2) later detected in individuals not reporting travel. Note that RDT-positive inbound passengers were excluded from this analysis, as they received treatment at the point of origin and thus, presumably did not import infections into the study area. The proportions of *csp* and *ama1* haplotypes meeting candidate importation criteria were compared to null distributions generated by permuting individual travel designations (i.e. whether participants reported travel) 1000 times. P-values were determined empirically from these null distributions.

### Malaria transmission modeling

We estimated the reproductive number (R) in Central Turkana using a modified Wallinga Teunis model [30] with varying stringency on genetic relatedness between infections; R values were computed for transmission networks based on 1) PCR-positivity alone, or 2) requiring individuals to share at least one *csp* or *ama1* haplotype. We also computed transmission networks based on infection with a specific haplotype.

To calculate R, we first determined transmission weights, *w*, for all infection pairs *(i,j)* where infection *i* was detected on the same day or later than infection *j* ($t_i \geq t_j$). In this analysis, we included only community participants for whom we had PCR data for all members of the household and RDT-negative/PCR-positive inbound passengers. Transmission weights were determined based on (a) time between infection detection ($w_t$) and (b) the geographic distance between infections at the facility level ($w_d$) and were the product of these two terms:

$$w(i,j) = w_t(i,j) \times w_d(i,j) \tag{1}$$

$w_t(i,j)$ was determined based on modeled serial interval distributions for treated symptomatic *P. falciparum* infections by Huber et al. [39], which follow a gamma distribution with parameters specified in Eq 2. Because the reactive case detection study design resulted in household member sampling within a few days of an index case, the interval between these infections was artificially shortened, resulting in low $w_t$ values. Thus, $w_t$ was conditioned on whether individuals reside in the same household, with maximum $w_t$ values assigned to household members based on the assumption that within-household transmission was highly likely:

$$w_t(i,j) = \begin{cases} \Gamma\left(t_i - t_j \,|\, 22,\ 2.1\right) & \text{Household } i \neq \text{Household } j \\ \max[\Gamma(22,\ 2.1)] & \text{Household } i = \text{Household } j \end{cases} \tag{2}$$

where $t_i - t_j$ was the time interval in days between infection *i* detection and infection *j* detection.

$w_d(i,j)$ was determined as previously described [31] based on the estimated distance between infections *i* and *j*:

$$w_d(i,j) = e^{-3d} \tag{3}$$

where *d* is the Haversine distance in km between health facilities to which index cases reported (for index and household members) or inbound passenger destinations. The decay factor of 3

was chosen as this distribution was previously observed in studies of *Anopheles* movement [31, 40, 41].

Reproductive numbers were determined by incorporating these time- and distance-based transmission weights into a Wallinga Teunis model [30]. First, the relative likelihood *p(i,j)* that case *i* was infected by case *j* was calculated based on the transmission weights described above:

$$p(i,j) = w(i,j) / \sum_{k \,|\, t_i \geq t_k} w(i,k) \tag{4}$$

Next, the effective reproductive number for case *j*, $R_j$, was calculated as the sum of *p(i,j)* over all *i* such that $t_i \geq t_j$:

$$R_j = \sum_i p(i,j) \tag{5}$$

The effective reproductive number, *Rt*, was calculated as the mean $R_j$ for all individuals at a given time *t*, smoothed over one-week time windows. The summary reproductive number *R* was calculated as the average of all $R_j$, excluding the first and last month of data collection. Confidence intervals (95%) were determined by bootstrap resampling 1000 times.

We estimated 3 distinct transmission networks with differing definitions of relatedness between pairs. Only PCR-positive community participants for whom we had PCR data for all members of the household (1014 index cases, 978 household members) as well as inbound passengers who were RDT-negative (not treated) and PCR-positive (n = 98) were included in the transmission network analyses (S10 Fig). For the transmission network of malaria-infected individuals, all pairs of PCR-positive infections within this subset were included in the analysis. More stringent transmission networks informed by parasite haplotypes required individuals to share at least one *csp* or one *ama1* haplotype. Finally, for haplotypes observed in at least twelve infections and present in both travelers and non-travelers, we also constructed haplotype-specific transmission networks (S3 Table).

For each transmission network, we estimated the contribution of travel to *P. falciparum* transmission by removing individuals who reported travel and comparing summary R values to corresponding networks that included all individuals using Bonferroni-corrected Wilcoxon Rank-Sum test.

## Data analysis and visualization

All statistical analyses and visualizations were performed in R (version 4.1.0) using the following packages: tidyverse (version 1.3.1) [42], rstatix (version 0.7.0) [43], geosphere (1.5–14) [44], EpiEstim (2.2–4) [45], igraph (1.2.7) [46], Hmisc (4.6–0) [47], tidygeocoder (1.0.5) [48], lubridate (1.8.0) [49], ggpubr (0.4.0) [50], scales (1.1.1) [51], ggrepel (0.9.1) [52], irr (0.84.1) [53], exactx2 (1.6.6) [54], flextable (0.6.9) [55], gtsummary (1.5.0) [56], infer (1.0.0) [57], ggridges (0.5.3) [58], ggraph (2.0.5) [59], tidygraph (1.2.0) [60].

## Supporting information

**S1 Fig. Map of Kenya with county borders.** Turkana County is outlined in orange. County fill color denotes the number of trips to or from the designated county. The Kitale bus terminal and Eldoret airport are marked with white points. The study area in Central Turkana is highlighted with an inset map of health facilities to which index cases reported. Shapefile for the map of Kenya and county borders obtained from the Humanitarian Data Exchange (HDX), an open platform for sharing data across crises and organizations: https://data.hum

data.org/dataset/cod-ab-ken. Inset map was created in QGIS (v 3.6.2-Noosa. Free and Open Source. QGIS Association. http://www.qgis.org.
(TIF)

**S2 Fig. Summary of travel histories stratified by case type.** (a) Number of trips reported by travelers. Note that survey limited the number of trips from the two months prior to enrollment to 3, and the number of trips for inbound passengers includes the trip that led to enrollment. Points represent median values. (b) Proportion of trips and residences (of inbound passengers) with *P. falciparum* parasite rate in children ages 2–10 (PR(2–10)) in 2018–2019 greater than that of Turkana. PR(2–10) obtained from the Malaria Atlas Project [26]. Trips to/within Turkana excluded.
(TIF)

**S3 Fig. *P. falciparum* incidence, parasite density, and multiplicity of infection in index cases, household members, and inbound passengers.** (a) PCR-positive (colored) and negative (grey) participants by study month and case type. (b) Parasite density distribution by case type. Points on x-axis represent median parasitemias for each case type. (c) Multiplicity of infection determined by *csp* (left) and *ama1* (right) haplotype counts in each individual. Points on x-axis represent median MOIs for each case type.
(TIF)

**S4 Fig. Distribution of haplotypes among individuals stratified by travel reported (purple) or not reported (blue), case type, and setting.**
(TIF)

**S5 Fig. *ama1* haplotype distribution and sharing in Central Turkana.** (a) A majority of *ama1* haplotypes were detected at multiple sites across Central Turkana. Columns are individual *ama1* haplotypes, and colors indicate the proportion of samples harboring the haplotypes detected at each of 6 enrollment sites. (b) The proportion of infection pairs sharing at least one *ama1* haplotype decreases with increasing time between infections. Each dot represents the proportion of all pairs separated by the indicated interval (with n = the color) that shared at least 1 *ama1* haplotype. (c) A marginal decrease in the proportion of individuals sharing at least one *ama1* haplotype with increasing distance (facility level) between infections was observed. Each dot represents the proportion of all pairs separated by the indicated geographic distance (n = the color) that shared at least 1 *ama1* haplotype. (d) The median proportion of individuals sharing at least one haplotype was greater within the household than outside the household. Within household, each dot represents the proportion of household members with whom an individual shared at least one haplotype. Outside household, each dot represents the median proportion of subsampled (n = number of household members, reps = 1000) infection pairs outside the household within ±60 days that share at least one haplotype.
(TIF)

**S6 Fig. *csp* haplotype distribution by site throughout the study period.**
(TIF)

**S7 Fig. *ama1* haplotype distribution by site throughout the study period.**
(TIF)

**S8 Fig. Candidate *csp* and *ama1* haplotypes imported into study area in Turkana, Kenya.** Seven *csp* haplotypes and 11 *ama1* haplotypes were potentially imported into the study area by 10 individuals. All potential importations occurred either through inbound passengers or

index cases reporting to the St. Patrick health facility.
(TIF)

**S9 Fig. Distribution of transmission weight inputs into Wallinga-Teunis model for (a) network of all PCR-positive infections, (b) network of individuals sharing at least one *csp* haplotype, (c) network of individuals sharing at least one *ama1* haplotype.**
(TIF)

**S10 Fig. Individuals included in transmission analysis for determining effective reproductive numbers (R and Rt).** Because household members were assigned maximum transmission weights in the transmission network analysis, only those community participants for whom we had qPCR data for all household members were included in the analysis. Those included in the transmission analysis (black outline) are shown as a subset of all samples collected.
(TIF)

**S11 Fig. Effect of reactive case detection on estimates of R.** To understand how the reactive case detection (RCD) format of this study may have impacted estimated R values, we simulated RCD in a previously published household-based longitudinal cohort in Webuye, Kenya [31] with varying limits on the number of households investigated per month. RCD was simulated by identifying RDT-positive symptomatic malaria infections (simulated index cases) and sampling household member samples from the next monthly visit closest in time to the index case (simulated reactive case detection). Although (a) there were differences in the overall estimated R between the full longitudinal cohort and the simulated, RCD datasets (Welch ANOVA $p < 0.0001$), no systematic differences were observed when RCD events per month were capped as they were in the present study. Additionally, there was good correlation between (b and c) Rt estimates for the full dataset and for simulated RCD (Spearman rho 0.794, $p < 0.0001$). Black line in (c) is line of identity.
(TIF)

**S1 Table. Baseline characteristics of index cases, household members, and inbound passengers.**
(DOCX)

**S2 Table. Characteristics of individuals carrying candidate imported haplotypes.**
(DOCX)

**S3 Table. Number of nodes in haplotype-specific transmission networks for *csp* and *ama1*.**
(DOCX)

**S1 Checklist. Inclusivity in global research checklist.**
(PDF)

## Acknowledgments

We thank all of the participants in and traveling to Central Turkana who gave their valuable time to participate in this study. We thank our field team, especially Dennis Okoth, Samuel Karanja, Elvis Ekitela, Jackson Kapelo, Rose Adome, Valentine Ramati, Naomi Eyanai, Sister Florence Wafula, and Ruth Areman.

## Author Contributions

**Conceptualization:** Diana Menya, Amy Wesolowski, Steve M. Taylor, Andrew A. Obala, Wendy P. O'Meara.

**Data curation:** Kelsey M. Sumner, Hannah R. Meredith.

**Formal analysis:** Christine F. Markwalter.

**Funding acquisition:** Steve M. Taylor, Andrew A. Obala, Wendy P. O'Meara.

**Investigation:** Elizabeth Freedman.

**Methodology:** Christine F. Markwalter, Amy Wesolowski, Elizabeth Freedman.

**Project administration:** Diana Menya, Daniel Esimit, Gilchrist Lokoel, Joseph Kipkoech, Lucy Abel, George Ambani.

**Supervision:** Diana Menya, Amy Wesolowski, Lucy Abel, George Ambani, Steve M. Taylor, Andrew A. Obala, Wendy P. O'Meara.

**Visualization:** Christine F. Markwalter.

**Writing – original draft:** Christine F. Markwalter.

**Writing – review & editing:** Diana Menya, Amy Wesolowski, Joseph Kipkoech, Elizabeth Freedman, Kelsey M. Sumner, Hannah R. Meredith, Steve M. Taylor, Andrew A. Obala, Wendy P. O'Meara.

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
