## [Decision Letter · Decision Letter 0]

7 Jun 2022

PGPH-D-22-00665

Importation of Plasmodium falciparum and sustained malaria transmission in a semi-arid region of Kenya

Dear Dr. Markwalter,

Thank you for submitting your manuscript to PLOS Global Public Health. After careful consideration, we feel that it has merit but does not fully meet PLOS Global Public Health’s publication criteria as it currently stands. Therefore, we invite you to submit a revised version of the manuscript that addresses the points raised during the review process.

Please submit your revised manuscript by . If you will need more time than this to complete your revisions, please reply to this message or contact the journal office at globalpubhealth@plos.org. Please include the following items when submitting your revised manuscript:

We look forward to receiving your revised manuscript.

Kind regards,

Louisa Alexandra Messenger, MSc, PhD

Academic Editor

Journal Requirements:

1. Please include a complete copy of PLOS’ questionnaire on inclusivity in global research in your revised manuscript. Our policy for research in this area aims to improve transparency in the reporting of research performed outside of researchers’ own country or community. The policy applies to researchers who have travelled to a different country to conduct research, research with Indigenous populations or their lands, and research on cultural artefacts. The questionnaire can also be requested at the journal’s discretion for any other submissions, even if these conditions are not met.  Please find more information on the policy and a link to download a blank copy of the questionnaire here: https://journals.plos.org/plosone/s/best-practices-in-research-reporting. Please upload a completed version of your questionnaire as Supporting Information when you resubmit your manuscript.

2. Please update your Competing Interests statement. If you have no competing interests to declare, please state: “The authors have declared that no competing interests exist.”

3. Please provide separate figure files in .tif or .eps format only and remove any figures embedded in your manuscript file. Please also ensure that all files are under our size limit of 10MB.

4. We have noticed that you have uploaded Supporting Information files, but you have not included a list of legends. Please add a full list of legends for your Supporting Information files after the references list.

Additional Editor Comments (if provided):

Reviewers' comments:

Reviewer's Responses to Questions

**Comments to the Author**

1. Does this manuscript meet PLOS Global Public Health’s publication criteria? Is the manuscript technically sound, and do the data support the conclusions? The manuscript must describe methodologically and ethically rigorous research with conclusions that are appropriately drawn based on the data presented.

Reviewer #1: Yes

Reviewer #2: Yes

Reviewer #3: Yes

2. Has the statistical analysis been performed appropriately and rigorously?

Reviewer #1: Yes

Reviewer #2: Yes

Reviewer #3: Yes

3. Have the authors made all data underlying the findings in their manuscript fully available (please refer to the Data Availability Statement at the start of the manuscript PDF file)?

Reviewer #1: Yes

Reviewer #2: Yes

Reviewer #3: No

4. Is the manuscript presented in an intelligible fashion and written in standard English?

Reviewer #1: Yes

Reviewer #2: Yes

Reviewer #3: Yes

5. Review Comments to the Author

Reviewer #1: The manuscript is well written and presented. It brings out an important aspect that needs to be addressed in many areas that are approaching elimination. However, there are minor comments / clarifications that need to be addressed. My comments are made in the submitted MS (attached).

Reviewer #2: The author has well established the population structure of parasites circulating in Turkana county in Kenya.

However, the following concerns need to be addressed.

1. The data as presented is sufficient in showing parasite diversity in Turkana but there is no evidence to associate transmission with recent travel from neighboring areas of high transmission.

2. Figure 2 (b), shows low positivity rates in inbound passengers and thus cannot warranty imply malaria importation into Turkana County. There is possibility of inbound traveler getting malaria infection while in Turkana.

3. The mathematical model used herein to represent importation and transmission rates fails to consider pertinent variables like the traveler's malaria infection history, traveler's origin, whether the traveler got infection while in Turkana.

4. Turkana is a marginalized area where the inhabitants practice nomadic lifestyle, the author fails to establish how their lifestyle can also contribute in the importation of malaria parasites from neighboring counties.

5. The factor of climate change, vector abundance, and congruence of malaria parasites with other zoonotic protozoa needs to be considered in the arguments.

In Conclusion, the issue of malaria importation should not be over emphasized but let the author report on parasite diversity/population structure of parasites in Turkana county as well as can further look at presence of parasite genes for drug resistance in the area.

Reviewer #3: Reviewer comments.

In this manuscript, the authors described the transmission pattern (local and imported cases) of malaria in a region of Kenya.

The manuscript shows that malaria transmission is sustained by local parasites, not by imported. The information provided in the manuscript could be used to guide interventions.

Globally, the manuscript is well-structured. However, the following comments and questions need be addressed.

1. Tableau: the “N” for sample size should be “n” as they sampled from the population. “N” would mean the entire population.

2. Figure 3. There are some haplotypes that are unique to specific enrollment sites. For instance, the haplotype H55, H63, and H60 are only seen in St. Patrick. What is the particularity of this site? This should be discussed in the manuscript.

3. Page 8, line 212: figure 4c is listed before figure 4a and 4b. This should be corrected. The figure 4c should be figure 4a.

4. Throughout the manuscript, Central Turkana and Turkana are used interchangeably. Are they the same? If yes, please use only one as it is confused.

5. Even though, the enrollment sites are well-described in the method section, a map in the supplemental material indicating these sites would be more informative for readers.

6. Make the ama1, csp sequence data available through a public repository or as a fasta file.

6. PLOS authors have the option to publish the peer review history of their article (what does this mean?). If published, this will include your full peer review and any attached files.

**Do you want your identity to be public for this peer review?** For information about this choice, including consent withdrawal, please see our Privacy Policy.

Reviewer #1: **Yes: **Abhinav Sinha

Reviewer #2: **Yes: **Nyamongo Bw'Onkoba

Reviewer #3: No

---

## [Editor Report · Decision Letter 1]

19 Jul 2022

Plasmodium falciparum importation does not sustain malaria transmission in a semi-arid region of Kenya

PGPH-D-22-00665R1

Dear Dr Markwalter,

We are pleased to inform you that your manuscript 'Plasmodium falciparum importation does not sustain malaria transmission in a semi-arid region of Kenya' has been provisionally accepted for publication in PLOS Global Public Health.

Best regards,

Louisa Alexandra Messenger, MSc, PhD

Academic Editor
